# Intra- and inter-individual variability in the underwater pull-out technique in 200 m breaststroke turns

Tomohiro Gonjo[1]*, Bjørn Harald Olstad[2], Jan Šťastný[3], Ana Conceição[4,5], Ludovic Seifert[6,7]

**1** Department of Rehabilitation & Sport Sciences, Bournemouth University, Poole, United Kingdom, **2** Department of Physical Performance, Norwegian School of Sport Sciences, Oslo, Norway, **3** Centre of Sports Activities, Brno University of Technology, Brno, Czech Republic, **4** Sport Sciences School of Rio Maior, Rio Maior, Portugal, **5** Research Centre of Sports, Health and Human Development (CIDESD), Rio Maior, Portugal, **6** CETAPS EA3832, Faculty of Sport Sciences, University of Rouen Normandie, Mont Saint Aignan, France, **7** Institut Universitaire de France (IUF), Paris, France

* tgonjo@bournemouth.ac.uk

**Data Availability Statement:** All relevant data are within the paper and its Supporting Information files.

## Abstract

The purpose of the present study was to investigate the intra- and inter-individual variability in arm-leg coordination during the underwater phase of the turn segment in 200 m breaststroke. Thirteen male swimmers were recruited and performed a 200 m breaststroke in a pre-calibrated 25 m pool. Sub-phases during the underwater segment were obtained using a notational analysis, and the mean velocity, displacement and duration during each sub-phase were obtained. A hierarchical cluster analysis (HCA) was performed using the analysed variables in all phases to identify inter-individual variability and random intra-individual variability. In addition, a linear mixed model (LMM: lap as a fixed effect and the participant as a random effect) was conducted to investigate systematic intra-individual variability. HCA identified three coordination patterns that were distinguished by the timing of the dolphin kick relative to the arm pull-out and the duration of the glide with arms at the side. All swimmers except one performed the arm pull-out after the dolphin kick. Nine swimmers maintained one coordination pattern, but other swimmers switched their coordination during the trial, particularly by shortening the duration of the glide with arms at the side. LMM showed a linear decrease (from the first to the last turn) in the time gap between the end of the dolphin kick and the start of the arm pull-out (a glide with the streamlined body position; F = 9.64, p = 0.034) and the glide duration with the arms at the side (F = 11.66, p = 0.015). In conclusion, both inter- and intra-individual variabilities during the underwater phase were evident in 200 m breaststroke turns, which were categorised into three patterns based on the timing of the dolphin kick and the duration of glides.

## Introduction

In competitive swimming races, forward movement can be divided into surface and underwater swimming, and both phases involve complex inter-limb (or inter-segment) coordination

**Funding:** AC received Portuguese Foundation for Science and Technology, I.P. under Grant UID04045/2020, JS obtained the Institutional developing project under grant RP 902025009 of Brno University of Technology, Czech Republic, and LS was supported by the French National Agency of Research under Grant ANR-19-STHP-0004, NePTUNE project. All funders were not involved in the study design, data collection and analysis, decision to publish or preparation of the manuscript.

**Competing interests:** The authors have declared that no competing interests exist.

[1, 2]. Inter-limb coordination during surface swimming in all four competitive swimming strokes and the body wave during underwater swimming have been widely investigated [3–7]. However, knowledge of the underwater phase in breaststroke swimming is less evident compared with other locomotive techniques in swimming. Swimmers perform several upper and lower limb motions during the breaststroke underwater phase after the start and turns, such as the arm pull to the thigh (arm pull-out), the forward motion of the arms and legs, one breaststroke kick and a part of the first breaststroke arm pull motion. Furthermore, under the current rule, a single dolphin kick is permitted after the start and each turn at any time before the first breaststroke kick (World Aquatics competition regulations 7.1 [8]). Therefore, swimmers should select the best order of the motion sequences and coordinate them well to minimise the deceleration and maximise the acceleration during the underwater phase.

Some researchers have conducted studies specifically focusing on the timing of the dolphin kick during the breaststroke underwater phase. For example, Hayashi et al. [9] investigated the optimal timing of the dolphin kick in relation to the arm pull-out motion using the SWUM simulation model [10] and reported that the optimal timing of the dolphin kick was 0.4 s before the initiation of the arm pull-down motion. However, the existence of the optimal motion is somewhat questionable in human motion because, according to the ecological dynamics framework, movement variability can play a functional role in allowing athletes to flexibly adapt their motion to a variety of constraints to achieve the objective of the task and to maximise the performance [11]. In other words, movement variability can be considered functional, rather than undesirable movement noises, when athletes show different movement patterns or coordination structures without any deterioration of the performance outcome [12]. A recent study [13] analysed the coordination between the upper and lower limbs during the underwater phase in 100 m breaststroke. They reported that there were three coordination patterns, even though all swimmers started their dolphin kick before the arm pull-out motion. More specifically, the three patterns were determined by the time gap between the dolphin kick and the following arm pull-out motion rather than the timing and coordination of other motions, such as arm and leg recoveries and propulsive movements. Another finding of the study was that two swimmers showed different coordination patterns in the start and turn phases, while others maintained a constant pattern. This result reasonably conforms to the ecological dynamics framework, as the swimmers are exposed to different environmental (e.g. flow velocity relative to the body), task (such as the individual pacing strategy and initial velocity) and organismic (e.g. fatigue and anthropometry) constraints [12, 14, 15] in the start and turn segments.

In the abovementioned study [13], all tested swimmers maintained their specific coordination pattern in the three turns during a short course 100 m breaststroke. However, it is unknown whether this is also the case in a 200 m race where the turn motion is repeated more than twice compared with a 100 m race. In a long course 200 m breaststroke race, the underwater phase distance decreases by 1 m in the last turn segment compared with the first one [16]. On the other hand, in a short course 100 m breaststroke (which has the same number of turns as long course 200 m races), the underwater distance in the last turn is shorter by only 0.5 m compared with the first turn [17]. Given that a change in the underwater distance exhibited in the 200 m breaststroke is twice larger than in the 100 m breaststroke race, it is possible that swimmers would exhibit greater inter-lap (intra-individual) arm-leg coordination variability during the underwater phase in the 200 m breaststroke turn segment than in the 100 m.

Another recent study [18] investigated the duration of arm-leg coordination during the start phase in 50 m, 100 m and 200 m breaststroke to assess the effect of the different race distances. They observed no difference in arm-leg coordination between the events but found slower speed and longer absolute duration and distance during the total glide phase (mainly

composed of the initial glide with the streamlined position and the glide with the arms at the side during the arm pull-out) in longer race events. These results imply that, depending on the race distance or the initial speed they obtain from the start, swimmers only adjust the least complex gliding duration but do not change complex inter-limb coordination to maximise their performance. However, as the study focused only on the start phase, it is unclear whether swimmers use the same strategy during the turn phase that involves accumulating fatigue during the race.

In summary, inter-limb coordination in the underwater phase during breaststroke turn segment should be investigated in line with the dynamic system theory because the coordination variation does not only explain the performance optimisation but also involves different constraints. Even though the literature suggests that swimmers generally maintain their individual coordination strategy, the inter-limb coordination during the underwater phase should be analysed in 200 m breaststroke as it would likely provide a better insight into the effect of fatigue on the coordination.

Therefore, the purpose of the present study was to investigate the intra- and inter-individual variability in arm-leg coordination in the underwater phase during the turn segment in 200 m breaststroke. It was hypothesised that the coordination of the swimmers would be classified into three patterns that are distinguished by the time gap between the dolphin kick and the following arm pull-out motion, as observed in the 100 m race [13], but they would switch the coordination pattern from one to another as the race progresses due to the fatigue effect. Obtaining detailed information on how swimmers manage their coordination patterns is essential as the foundation knowledge for researchers and practitioners to further consider how they are related to swimming performance.

## Methods

### Participants

Thirteen male competitive swimmers (18.4 ± 2.7 years age; 182.8 ± 2.7 cm height; and 76.6 ± 11.1 kg weight) specialised in breaststroke were recruited. Their personal best record in the short course 200 m breaststroke was 141.0 ± 6.7 s, which corresponded to 627.4 ± 93.3 FINA point and 85.4 ± 4.0% of the 200 m breaststroke world record. All participants were informed about the procedures, benefits and potential risks of the study orally and in writing, and thereafter, swimmers and their legal guardians (for minors) provided their written informed consent. The procedure of the current study was approved by the local ethical committee and the National Data Protection Agency for Research in accordance with the Declaration of Helsinki.

### Data collection

Data collection was conducted at an indoor 25 m swimming pool (27 ˚C water temperature, 29 ˚C air temperature and 55% humidity). Swimmers underwent their personal warm-up session on land and in water for about 60 minutes to ensure that each swimmer would have a similar physiological condition as in a real competition. After 20–30 min of active rest, when the swimmers changed to their competition swimsuits, they performed a 200 m breaststroke with the aim of achieving the fastest time possible. Each participant performed the time trial once in the centre lane of the pool without other competitors.

The 200 m time trial was recorded by the AIM race analysis system (AIMsys Sweden AB, Lund, Sweden) that was synchronised with an electronic Omega timing system (Swiss Timing, Bienne, Switzerland). The system consisted of five underwater and five on-land cameras with a 50 Hz sampling frequency, which covered the entire 25 m of the centre lane. The underwater

cameras and on-land cameras were located 0.7 m below and 5 m above the water surface, respectively, and the distance between each camera on the same level was approximately 5 m. The camera system was calibrated as described in Haner et al. [19], which yielded a mean error of 2.0 pixels, corresponding to a 0.025˚ angular and 0.006 m linear error. The system was equipped with an automatic yellow colour tracking function based on image processing and machine learning algorithms, which enabled automatic tracking of the centre of the swimmer's head with a yellow silicone cap. The combination of the camera calibration and the automatic head tracking generated the horizontal head displacement and velocity data throughout the time trial. Furthermore, footage from different cameras was all merged based on the global coordinate data that produced a panning video file of the entire 200 m time trial. Before the merging process, the image distortion and the aspect ratio of each footage were fixed as described by Haner et al. [19], and the exported video was used for the further qualitative analysis described below.

## Data analysis

As the primary focus of the present study was arm-leg motion during the breaststroke underwater phase after turns, the duration when swimmers performed active motion in the underwater phase (defined as the active underwater phase; from the first leg or arm motion after the push-off to the head breakout) was observed and analysed. The same approach as in a recent study [13] was used to quantify the duration and travelled distance during propulsive or resistive motions as well as during six time gaps using key sequences. The key sequences included the beginning and the end of the dolphin kick ($DK_{beg}$ and $DK_{end}$), arm pull-out ($APO_{beg}$ and $APO_{end}$), arm recovery ($AR_{beg}$ and $AR_{end}$) and leg recovery ($LR_{beg}$ and $LR_{end}$). The time gaps were obtained by subtracting the time or distance in one key sequence from the other (Table 1 and Fig 1), meaning that the time gap can be either positive or negative, depending on which

**Table 1. Definition of analysed active motions and time gaps in the present study.**

| | Name | Definition |
|---|---|---|
| | Initial glide | From the wall push-off until the first active action of arms or legs |
| Active Motion | Dolphin kick | From the beginning to the end of the downbeat |
| | Arm pull-out | From the beginning to the end of active hand backward motion during the pull-out motion in the underwater phase |
| | Recovery | The total duration of recovery motions (arm and leg recovery after the arm pull-out and the leg recovery motion for the dolphin kick) |
| Time gap | $APO_{end}$—$AR_{beg}$ [T1] | The time gap between the end of the arm pull-out and the beginning of the following arm recovery motion |
| | $LR_{beg}$—$AR_{beg}$ [T2] | The time gap between the beginning of the leg and arm recovery after the arm pull-out ($LR_{beg}$—$AR_{beg}$ [T2] < 0; the arm recovery started earlier than the leg recovery) |
| | $AR_{end}$—$LR_{end}$ [T3] | The time gap between the end of the arm and leg recovery before the first breaststroke kick and arm stroke ($AR_{end}$—$LR_{end}$ [T3] < 0; the legs have finished the recovery motion and started the breaststroke kick while the arms are still in recovery motion) |
| | $DK_{end}$—$APO_{beg}$ [T4] | The time gap between the end of the dolphin kick and the beginning of the arm pull-out ($DK_{end}$—$APO_{beg}$ [T4] > 0; the dolphin kick finished before the arms started performing the pull-out motion) |
| | $APO_{end}$—$DK_{beg}$ [T5] | The time gap between the end of the arm pull-out and the beginning of the dolphin kick ($APO_{end}$—$DK_{beg}$ [T5] < 0; the beginning of the dolphin kick occurred before the end of the arm pull-out) |
| | $DK_{end}$—$LR_{beg}$ | The time gap between the end of the dolphin kick and the beginning of leg recovery |

key sequence occurs first. For example, in the case of $DK_{end}$—$APO_{beg}$, this time gap shows positive when the $DK_{end}$ is performed before $APO_{beg}$ and exhibits negative when $APO_{beg}$ occurs first. Among the six time gaps, five time gaps were the same as those investigated in the literature [13, 18]. Therefore, those time gaps were also assigned sub-names (T1-T5) for consistency.

All key sequences were detected qualitatively, i.e. with visual observation of video footage by five investigators using the blind technique. The five members had at least two-year experience in swimming coaching or/and technical support at the national or international level. Before the qualitative analysis for the 200 m turns, all five operators obtained the key moments from one swimmer's 100 m breaststroke trial (three turns) as a preliminary analysis. An intraclass correlation (ICC) was then computed to assess inter-operator variability for each key moment, which yielded ICC > 0.999 with p < 0.001, and the maximum inter-investigator difference in all key moments in the 200 m turn analysis was 0.04 s. See Seifert et al. [13] for a detailed process of qualitative analysis.

The mean velocity during the active underwater phase was calculated by dividing the travel distance (quantified by the head displacement) by the time spent during this phase. The duration and distance during each active motion or time gap were derived as the absolute value and the relative value in relation to the total active underwater duration or distance. The total time and the first 15 m time in each lap were also obtained using the head data obtained by the AIM system and the time data derived from the electronic Omega timing system.

## Statistical and machine learning analyses

To analyse inter-individual variability and random intra-individual variability, a hierarchical cluster analysis with the Euclidean distance dissimilarity measure and the complete linkage method was used [20, 21]. The cluster analysis was performed using 18 variables (relative duration and distance of the variables presented in Table 1, except for the initial glide that is not relevant to the arm-leg coordination), and each variable contained 91 samples (13 swimmers × seven turns). All variables were normalised using the Z-score standardisation so that all variables had comparable scales. The number of clusters that best explained the data was validated using the Calinski-Harabasz (CH) index [22], and the contribution of each variable to the clustering was assessed using Fisher information, which is the ratio of inter-cluster to intra-cluster distances. In other words, Fisher information of > 1 means that the inter-cluster distance is greater than the intra-cluster distance, implying that the variable contributed to the clustering to a large extent. The CH index was calculated for two to ten potential clusters with Cluster Validity Analysis Platform (CVAP, Version 3.7) [23] designed for MATLAB (R2014a, 1994–2014, MathWorks Inc, Massachusetts).

To investigate systematic intra-individual variability, the travelled distance, duration and velocity during the active underwater phase were assessed with a linear mixed model with the lap as a fixed effect and the participant as a random effect. The same method was also used to investigate the systematic intra-individual variability in the relative duration and distance in each limb motion and time gaps described in Table 1. Before the linear mixed model analysis, outliers in the dataset were excluded, and the normality of all data was checked and confirmed with the Shapiro-Wilk test. Due to the large number of analyses, $p$-values obtained from the linear mixed model analyses were corrected using Holm-Bonferroni procedure [24] to minimise the Type I error risk. The statistical significance was set at $p = 0.05$. The analyses using the linear mixed model were performed with IBM SPSS Statistics 24 (IBM Corporation, Somers, NY, USA).

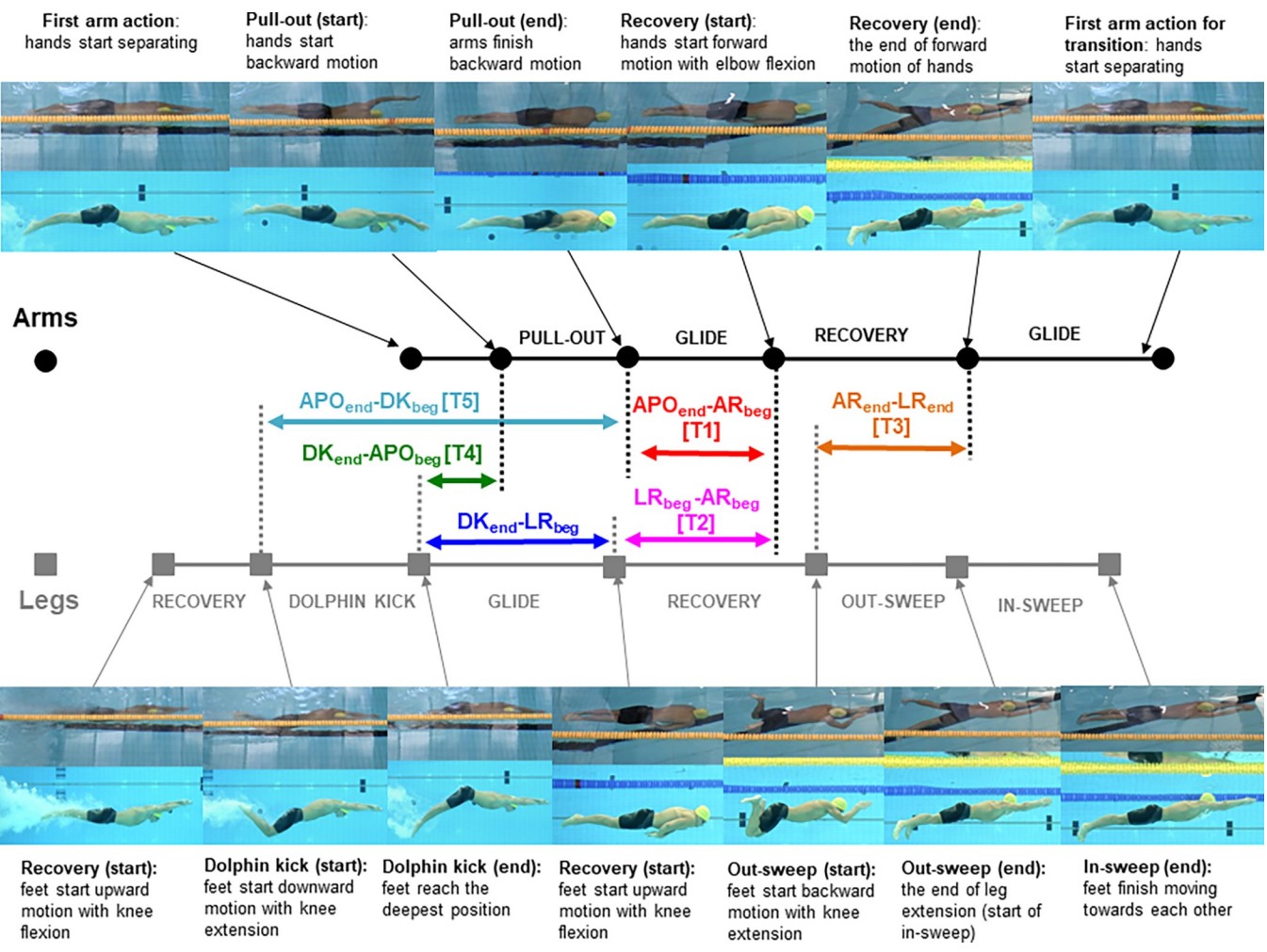

**Fig 1. Key movements and phases of arm and leg movements for the data analysis.**

## Results

The 200 m time trial result was 147.76 ± 8.04 s, which was 4.4% slower than their personal best record. The cluster analysis with CH index validation showed three clusters, meaning that the underwater motion of the 200 m breaststroke turn phase can be categorised into three patterns (C1, C2 and C3). Nine swimmers maintained the same pattern throughout the 200 m time trial, while four swimmers switched their underwater coordination pattern from C1 to C2 during the time trial (Fig 2). Among the nine swimmers who maintained the same pattern, six, two and one swimmers showed C1, C2 and C3 pattern, respectively. The absolute time and duration of each time gap as well as 15 m and 25 m time for each cluster are presented in Table 2. When normalising the 15 m time by the 25 m time, the 15 m time for C1, C2 and C3 corresponded to 58.04 ± 0.81%, 58.69 ± 0.93% and 57.31 ± 0.43% of the 25 m time, respectively.

The mean relative duration of the dolphin kick, arm pull-out and recovery motion, as well as each time gap for each cluster, are exhibited both in Table 3 and Fig 3. Fisher information (Table 3) demonstrated that the three patterns were mainly distinguished by $APO_{end}$—$AR_{beg}$ [T1], $DK_{end}$—$APO_{beg}$ [T4], $APO_{end}$—$DK_{beg}$ [T5] and $DK_{end}$—$LR_{beg}$, which had Fisher

**Fig 2. Coordination patterns observed during each turn for all swimmers.**

**Table 2. Mean (SD) of 15 m time, 25 m time, and absolute time and displacement of the active underwater phases in each cluster.**

| | Phase | C1 (n = 59) | | C2 (n = 25) | | C3 (n = 7) | |
|---|---|---|---|---|---|---|---|
| Duration (s) | 15 m | 10.97 | (0.64) | 11.36 | (0.78) | 10.51 | (0.29) |
| | 25 m | 18.97 | (1.79) | 19.36 | (1.35) | 18.33 | (0.38) |
| | Dolphin kick | 0.20 | (0.04) | 0.20 | (0.03) | 0.19 | (0.01) |
| | Arm pull-out | 0.64 | (0.09) | 0.65 | (0.12) | 0.75 | (0.07) |
| | Recovery | 1.21 | (0.16) | 1.21 | (0.12) | 0.97 | (0.10) |
| | APOend—$AR_{beg}$ [T1] | 0.66 | (0.18) | 0.41 | (0.12) | 0.75 | (0.25) |
| | $LR_{beg}$—$AR_{beg}$ [T2] | -0.35 | (0.20) | -0.34 | (0.18) | -0.14 | (0.14) |
| | $AR_{end}$—$LR_{end}$ [T3] | -0.14 | (0.04) | -0.14 | (0.07) | -0.16 | (0.02) |
| | $DK_{end}$—$APO_{beg}$ [T4] | 0.16 | (0.19) | 0.08 | (0.08) | -0.72 | (0.07) |
| | $APO_{end}$—$DK_{beg}$ [T5] | -1.02 | (0.20) | -0.87 | (0.17) | -0.21 | (0.02) |
| | $DK_{end}$—$LR_{beg}$ | 1.88 | (0.29) | 1.47 | (0.17) | 0.92 | (0.18) |
| Distance (m) | Dolphin kick | 0.32 | (0.07) | 0.30 | (0.04) | 0.41 | (0.02) |
| | Arm pull-out | 1.17 | (0.13) | 1.17 | (0.12) | 1.40 | (0.12) |
| | Recovery | 1.07 | (0.17) | 1.16 | (0.17) | 0.96 | (0.16) |
| | APOend—$AR_{beg}$ [T1] | 1.06 | (0.27) | 0.66 | (0.20) | 1.33 | (0.42) |
| | $LR_{beg}$—$AR_{beg}$ [T2] | 0.44 | (0.22) | 0.46 | (0.24) | 0.23 | (0.21) |
| | $AR_{end}$—$LR_{end}$ [T3] | 0.12 | (0.05) | 0.13 | (0.05) | 0.16 | (0.03) |
| | $DK_{end}$—$APO_{beg}$ [T4] | 0.31 | (0.18) | 0.09 | (0.11) | 1.35 | (0.10) |
| | $APO_{end}$—$DK_{beg}$ [T5] | 1.70 | (0.27) | 1.60 | (0.15) | 0.46 | (0.04) |
| | $DK_{end}$—$LR_{beg}$ | 3.05 | (0.38) | 2.42 | (0.24) | 1.60 | (0.33) |

Values in italics show the median (inter-quartile range) for non-normal distribution data.

**Table 3. Mean (SD) relative time and displacement of the active underwater phases in each cluster and Fisher information of each variable.**

| | Phase | C1 (n = 59) | | C2 (n = 25) | | C3 (n = 7) | | Fisher information |
|---|---|---|---|---|---|---|---|---|
| Relative duration (%) | Dolphin kick | 5.66 | (1.72) | 6.07 | (1.14) | 6.03 | (0.56) | 0.05 |
| | Arm pull-out | 17.95 | (2.69) | 20.01 | (2.63) | 24.13 | (1.65) | 0.45 |
| | Recovery | 32.60 | (4.51) | 38.44 | (4.11) | 31.35 | (4.17) | 0.28 |
| | APOend—$AR_{beg}$ [T1] | 19.24 | (6.11) | 11.78 | (2.92) | 23.79 | (5.40) | 0.82 |
| | $LR_{beg}$—$AR_{beg}$ [T2] | -9.84 | (5.35) | -10.17 | (5.59) | -4.81 | (4.23) | 0.07 |
| | $AR_{end}$—$LR_{end}$ [T3] | -3.96 | (1.35) | -4.22 | (2.36) | -5.16 | (0.53) | 0.14 |
| | $DK_{end}$—$APO_{beg}$ [T4] | 5.03 | (6.98) | 2.18 | (1.91) | -23.28 | (1.60) | 3.03 |
| | $APO_{end}$—$DK_{beg}$ [T5] | -27.61 | (4.11) | -28.26 | (3.94) | -6.88 | (0.74) | 2.04 |
| | $DK_{end}$—$LR_{beg}$ | 51.09 | (6.80) | 44.14 | (4.77) | 29.45 | (3.47) | 0.97 |
| Relative distance (%) | Dolphin kick | 6.34 | (1.67) | 6.93 | (1.19) | 8.83 | (0.89) | 0.20 |
| | Arm pull-out | 23.96 | (3.62) | 27.08 | (3.29) | 30.35 | (2.34) | 0.33 |
| | Recovery | 21.72 | (3.55) | 26.90 | (3.64) | 20.85 | (3.85) | 0.41 |
| | APOend—$AR_{beg}$ [T1] | 22.91 | (7.21) | 14.23 | (3.25) | 28.36 | (6.27) | 0.84 |
| | $LR_{beg}$—$AR_{beg}$ [T2] | 9.36 | (4.90) | 8.54 | (7.04) | 5.13 | (4.24) | 0.06 |
| | $AR_{end}$—$LR_{end}$ [T3] | 2.54 | (1.11) | 3.30 | (1.15) | 3.40 | (0.47) | 0.12 |
| | $DK_{end}$—$APO_{beg}$ [T4] | 6.15 | (3.38) | 1.92 | (2.76) | 29.30 | (2.16) | 4.86 |
| | $APO_{end}$—$DK_{beg}$ [T5] | 34.68 | (4.36) | 36.49 | (4.90) | 9.88 | (0.94) | 2.44 |
| | $DK_{end}$—$LR_{beg}$ | 60.49 | (7.00) | 54.17 | (5.08) | 34.39 | (4.22) | 1.21 |

Values in italics show the median (inter-quartile range) for non-normal distribution data.

information close to or over 1.0 in both relative duration and distance. C3 differed from C1 and C2 in all these time gaps, but the difference was particularly evident in $DK_{end}$—$APO_{beg}$ [T4] and $APO_{end}$—$DK_{beg}$ [T5], both of which were related to the timing of the dolphin kick in relation to the arm pull-out. The difference between C1 and C2 was less evident in $DK_{end}$—$APO_{beg}$ [T4] and $APO_{end}$—$DK_{beg}$ [T5] (2% and 0.7% difference, respectively), but C1 had a longer relative duration and distance in APOend—$AR_{beg}$ [T1] and $DK_{end}$—$LR_{beg}$ (both related to the glide motion with arms at the side after the dolphin kick) compared with C2.

The relative duration and distance in APOend—$AR_{beg}$ [T1] and $DK_{end}$—$LR_{beg}$ for the four swimmers who switched their coordination patterns during the 200 m time trial are illustrated in Fig 4. In $DK_{end}$—$LR_{beg}$, two swimmers (swimmer 6 and 9) had a relative duration and distance around or above the C1 mean, one swimmer (swimmer 11) constantly showed a lower relative distance and duration than the C2 mean, swimmer 12 reduced the relative distance and duration from around the C1 mean to C2 mean as the trial progressed. In APOend—$AR_{beg}$ [T1], three swimmers reduced both duration and distance as the trial progressed. However, one swimmer (swimmer 6) had a random APOend—$AR_{beg}$ [T1] pattern. In five turns, swimmer 6 showed the relative distance and duration of APOend—$AR_{beg}$ [T1] around the C2 mean. However, in the third turn, swimmer 6 showed a much lower APOend—$AR_{beg}$ [T1] relative distance and duration than the C2 mean. On the contrary, swimmer 6 had a longer APOend—$AR_{beg}$ [T1] relative distance and duration than the C1 mean in the fourth lap.

In linear mixed modelling analyses, there was a significant negative lap effect on the relative distance in the initial glide phase (Table 4 and Fig 5), but no significant lap effect was found in the relative duration of the phase. Significant negative lap effects on the relative distance and duration of the active underwater phase were also observed (both $p < 0.001$, Table 4 and Fig 5). On the other hand, there was no significant lap effect in the active underwater velocity ($p = 0.06$). In all three motions during the active underwater phase (dolphin kick, arm pull-out

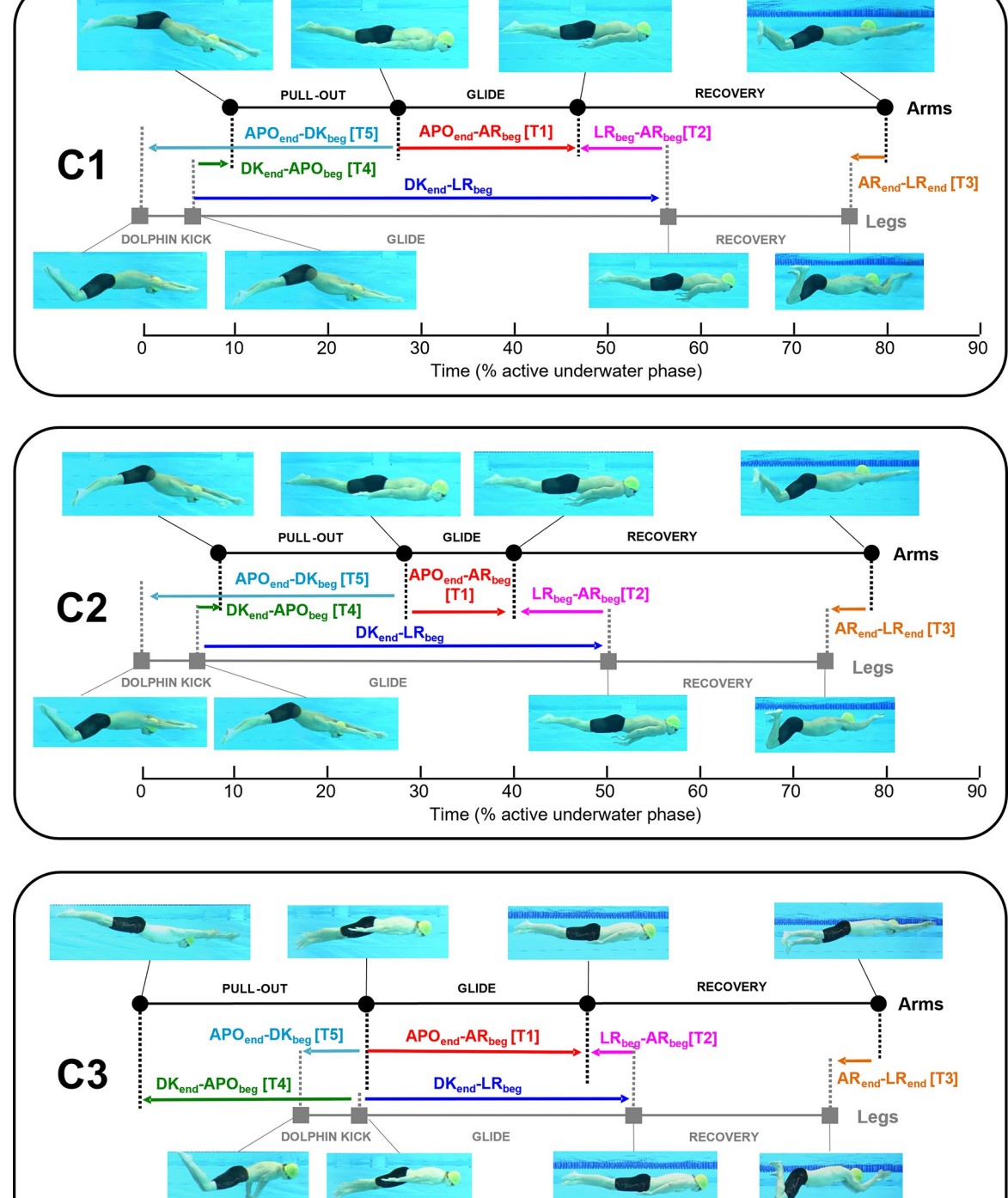

**Fig 3. Diagram of the relative duration of dolphin kick, arm pull-out and recovery motion as well as each time gap for each cluster.** Right-pointing and left-pointing arrows show the positive and negative time gap, respectively.

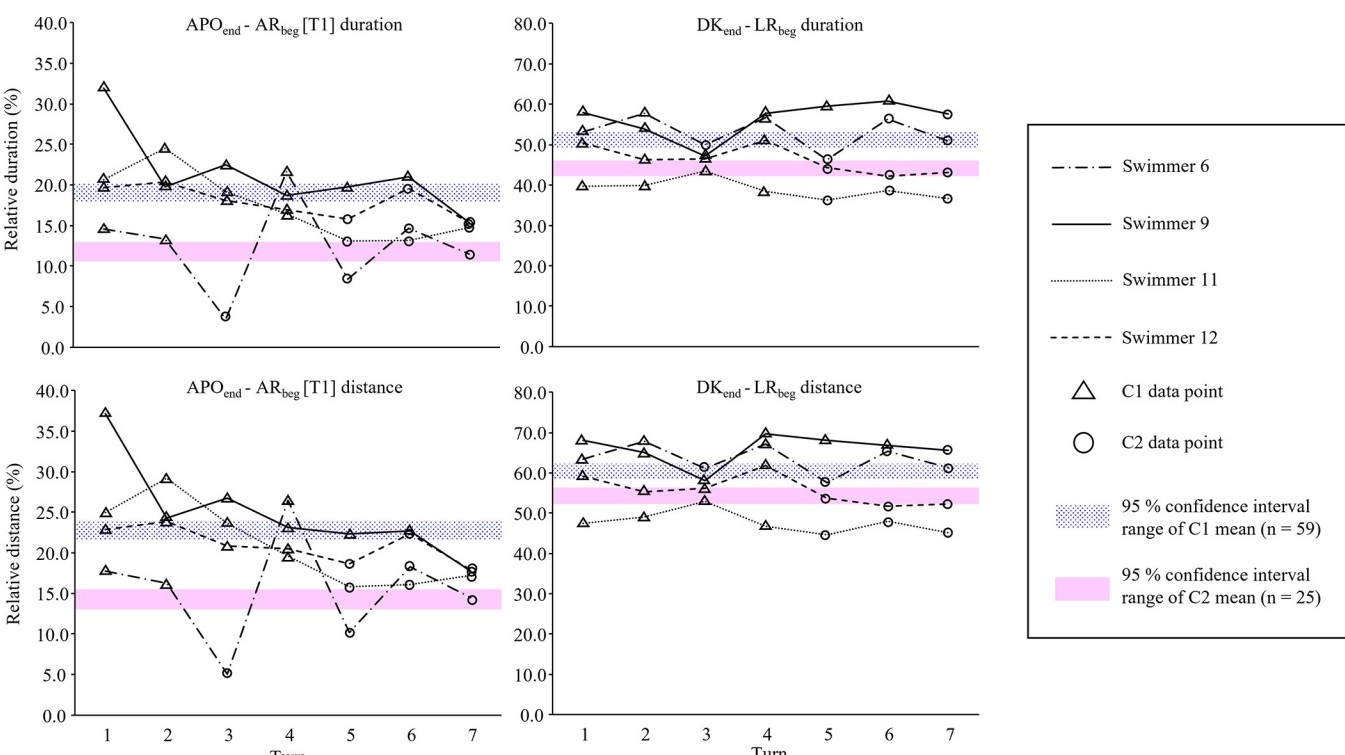

**Fig 4. The relative duration and distance in APO$_{end}$—AR$_{beg}$ [T1] and DK$_{end}$—LR$_{beg}$ for the four swimmers who switched the coordination pattern during the trial.**

and recovery), significant positive lap effects were observed in both relative duration and distance ($p \leq 0.005$, Table 4 and Fig 5).

Among the time gaps during the active underwater phase, significant lap effects were observed in both the relative duration and distance of APOend—AR$_{beg}$ [T1], DK$_{end}$—APO$_{beg}$ [T4] and APO$_{end}$—DK$_{beg}$ [T5], as well as the relative distance of APOend—AR$_{beg}$ [T1] and DK$_{end}$—LR$_{beg}$ (Table 5 and Fig 6). The lap effect was negative in all cases except for the relative distance of APO$_{end}$—DK$_{beg}$ [T5]. However, it should be noted that the result of the relative duration of APO$_{end}$—DK$_{beg}$ [T5], which had negative values in all laps, should be treated with

**Table 4. Results from the mixed linear modelling assessing the effect of lap on the velocity, distance and duration of the active underwater phase, and the relative duration and distance of each motion.**

| Phase | Variable | F-value | p-value |
|---|---|---|---|
| Initial glide | Duration (%) | 7.64 | 0.064 |
| | Distance (%) | 9.17 | **0.037** |
| Active underwater phase | Velocity (m·s⁻¹) | 7.26 | 0.060 |
| | Distance (m) | 58.95 | **< 0.001** |
| | Duration (s) | 33.71 | **< 0.001** |
| Dolphin kick | Duration (%) | 24.89 | **< 0.001** |
| | Distance (%) | 15.24 | **0.005** |
| Arm pull-out | Duration (%) | 48.03 | **< 0.001** |
| | Distance (%) | 38.29 | **< 0.001** |
| Recovery | Duration (%) | 19.53 | **< 0.001** |
| | Distance (%) | 16.05 | **0.003** |

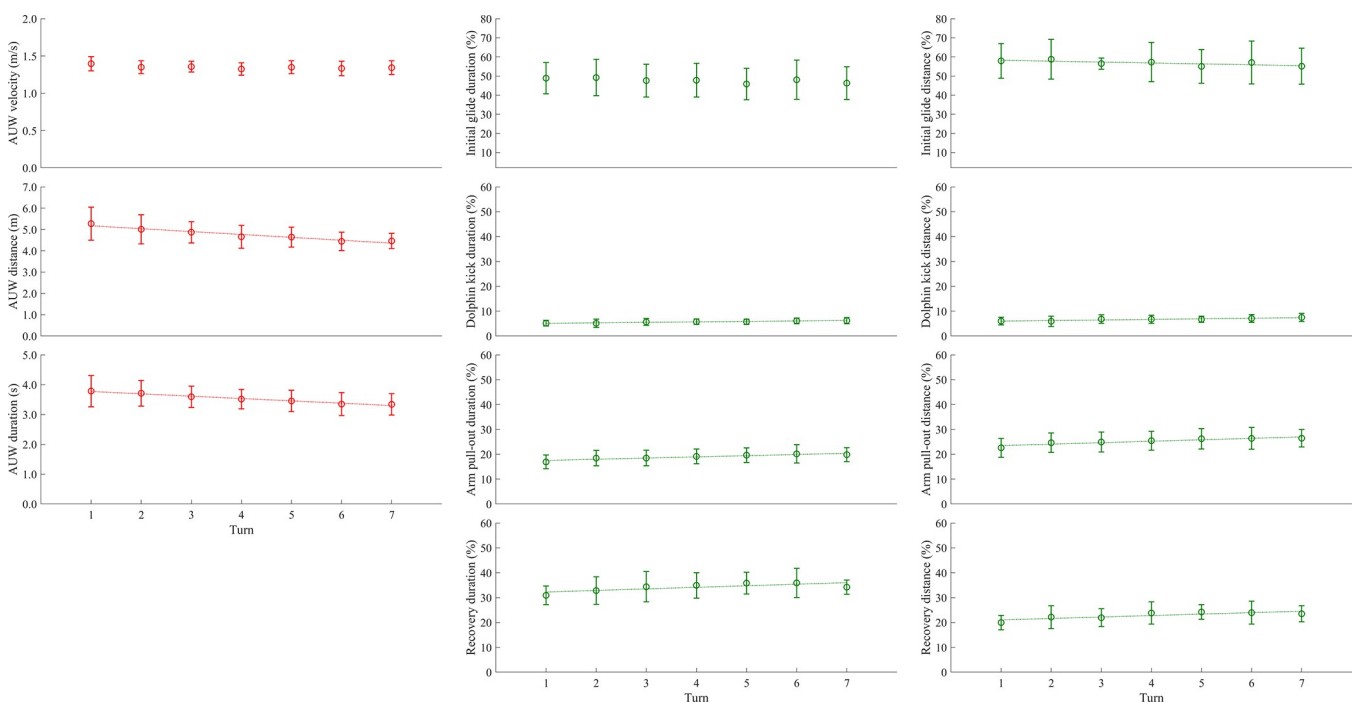

**Fig 5. Absolute active underwater (AUW) velocity, distance and duration, as well as the relative duration and distance in each limb motion.** Graphs with a linear trend line and a regression equation represent the variables that had a significant lap effect in the linear mixed model analysis.

caution because the positive and negative signs of the time gaps only indicate the order of the motion sequences in the present study (e.g. $APO_{end}$—$DK_{beg}$ [T5] < 0 means that the beginning of the dolphin kick is before the end of the arm pull-out, and $APO_{end}$—$DK_{beg}$ [T5] > 0 shows that the dolphin kick starts after the end of the arm pull-out), meaning that the increase or decrease in the duration should be interpreted without considering the sign. As the $APO_{end}$—$DK_{beg}$ [T5] relative duration linearly changed from -26.6% (first turn) to -28.8% (last turn), the relative duration increased by 2%. It should also be noted that in $DK_{end}$—$APO_{beg}$ [T4] and $APO_{end}$—$DK_{beg}$ [T5] relative duration and distance, data obtained from swimmer 10 were detected as outliers in all laps and therefore excluded.

**Table 5. Results from the mixed linear modelling assessing the effect of lap on the relative duration and distance of each time gap.**

| Phase | Variable | F-value | p-value |
|---|---|---|---|
| $APO_{end}$—$AR_{beg}$ [T1] | Duration (%) | 18.43 | **0.002** |
| | Distance (%) | 22.06 | **< 0.001** |
| $LR_{beg}$—$AR_{beg}$ [T2] | Duration (%) | 2.57 | 0.219 |
| | Distance (%) | 2.45 | 0.253 |
| $AR_{end}$—$LR_{end}$ [T3] | Duration (%) | 6.09 | 0.070 |
| | Distance (%) | 2.88 | 0.300 |
| $DK_{end}$—$APO_{beg}$ [T4] | Duration (%) | 9.64 | **0.034** |
| | Distance (%) | 24.24 | **< 0.001** |
| $APO_{end}$—$DK_{beg}$ [T5] | Duration (%) | 11.66 | **0.015** |
| | Distance (%) | 17.30 | **0.002** |
| $DK_{end}$—$LR_{beg}$ | Duration (%) | 7.64 | 0.055 |
| | Distance (%) | 9.40 | **0.036** |

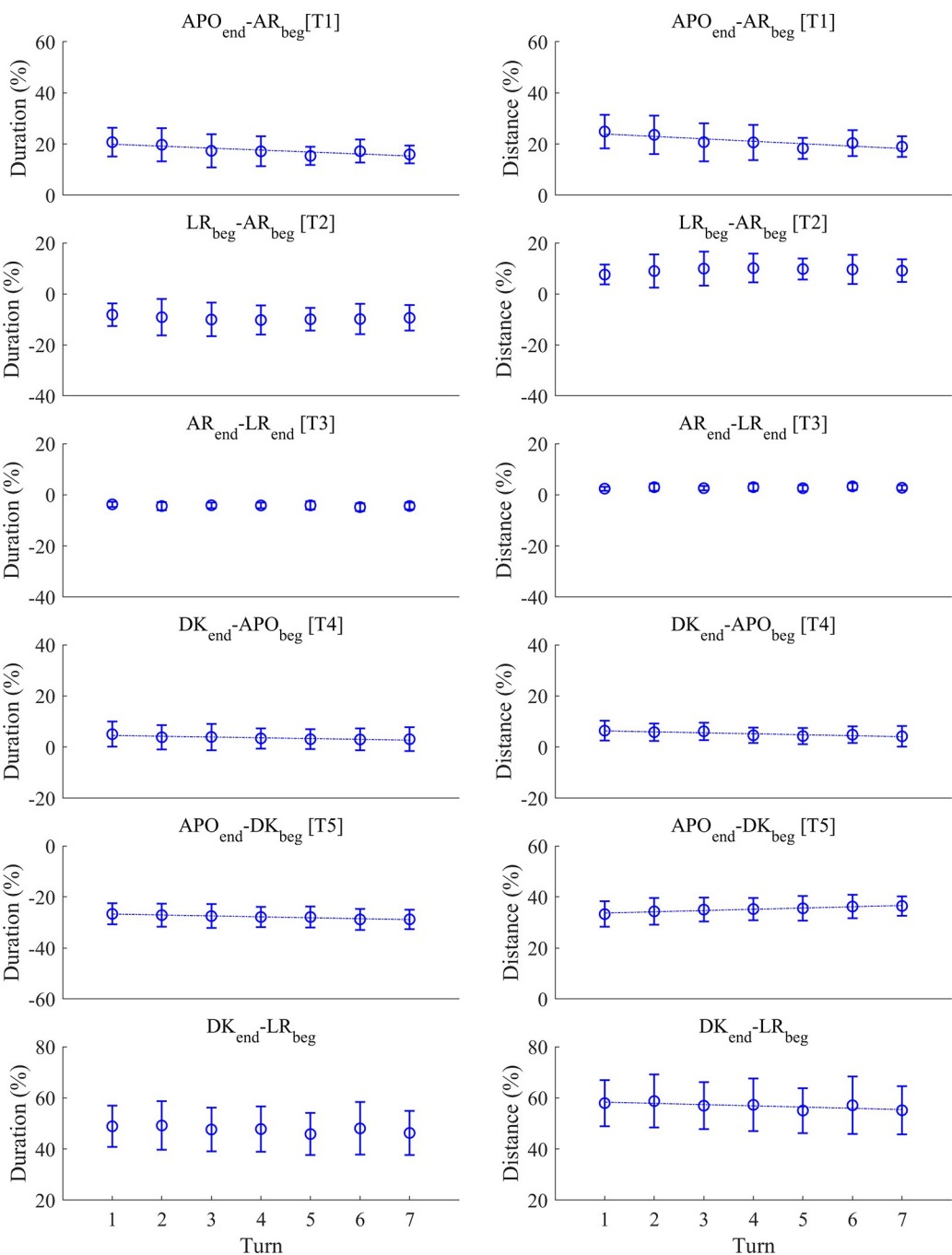

**Fig 6. The relative duration and distance in each time gap.** Graphs with a linear trend line and a regression equation represent the variables that had a significant lap effect in the linear mixed model analysis.

## Discussion

The purpose of the present study was to investigate the intra- and inter-individual variability in arm-leg coordination in the underwater phase during the turn segment in 200 m breaststroke, hypothesising that the coordination of the swimmers would be categorised into three patterns that are distinguished by the time gap between the dolphin kick and the following

arm pull-out motion. It was also hypothesised that swimmers would show different coordination patterns between the beginning and the end of the time trial due to the fatigue effect. Results showed that arm-leg coordination during the turn segment was categorised into three patterns (C1, C2 and C3). Four swimmers switched from one pattern to another in the 200 m time trial, but most swimmers maintained their coordination pattern. Therefore, the initial hypotheses were partially supported.

## Inter-individual variability

Although the current study found three coordination patterns, as was the case in a recent 100 m breaststroke study [13], the characteristics of the three patterns found in the present study were somewhat different from the previous study. In the previous study, the three patterns were distinguished by the timing of the dolphin kick in relation to the arm pull-out (performing the arm pull-out after the dolphin kick with a time gap, with an overlap or continuously). However, in the present study, the continuous pattern was not observed. Among the three coordination patterns (C1, C2 and C3) established in the present study, C1 and C2 showed $DK_{end}$—$APO_{beg}$ [T4] $> 0$, which means that they were both characterised by the arm pull-out being performed after the completion of the dolphin kick following a time gap. On the contrary, $DK_{end}$—$APO_{beg}$ [T4] $< 0$ in C3 suggested that the swimmer with this pattern (swimmer 10) started performing the dolphin kick before the completion of the arm pull-out.

Swimmer 10 maintained this pattern throughout the time trial. Considering that $APO_{end}$—$DK_{beg}$ [T5] in C3 was -6.9% (the beginning of the dolphin kick was around 6.9% time earlier than the end of the arm pull-out) and that the duration of the dolphin kick was 6.0% time (while the duration of the arm pull-out was 24.1%), it can be concluded that swimmer 10 coordinated the dolphin kick and the arm pull in such a way that the two motions ended almost at the same time. Assuming that the latter half of the arm pull-out is performed with elbow extension, as is the case in front crawl push motion [25], it is reasonable that the swimmer exhibited this type of coordination because synchronising the knee and elbow extension (in-phase coordination) is the simplest form of inter-limb coordination pattern [26].

Differences between C1 and C2 were clear in APOend—$AR_{beg}$ [T1] and $DK_{end}$—$LR_{beg}$. As shown in Fig 3, APOend—$AR_{beg}$ [T1] occurred during the $DK_{end}$—$LR_{beg}$ phase, which means that APOend—$AR_{beg}$ [T1] is the phase where both the arms and legs are in the glide phase (the glide with arms at the side). Given that the difference between C1 and C2 in APOend—$AR_{beg}$ [T1] was 8.6%, while the difference in $DK_{end}$—$LR_{beg}$ was 6.3%, the primary difference in $DK_{end}$—$LR_{beg}$ can, in fact, also be explained as the difference in the glide with arms at the side. Therefore, in general, C1 is characterised by a longer relative duration and distance in the glide motion with arms at the side compared with C2.

## Intra-individual variability

There were four swimmers who changed the underwater motion pattern from C1 to C2. In a study focusing on the short course 100 m breaststroke, a similar coordination change was observed from the start to turn segments, but swimmers maintained the same pattern in the three turns [13]. This implies that the constraints that caused the coordination change in 100 m and 200 m breaststroke are different. In 100 m breaststroke, swimmers probably change their coordination pattern due to differences in the initial speed, travel direction and flow velocity around the body between the start and turns, but in 200 m breaststroke, the observed change was likely due to fatigue [12, 14, 15].

Among the four participants who changed the coordination pattern, three swimmers showed a reduction in APOend—$AR_{beg}$ [T1] distance and duration, meaning that these three

swimmers changed their pattern by shortening the glide with arms at the side. However, the change in swimmer 6 cannot be explained by this phenomenon. Unlike the other three participants, swimmer 6 showed a random change in APOend—$AR_{beg}$ [T1] relative duration and distance. In the first two turns, even though this swimmer showed APOend—$AR_{beg}$ [T1] relative duration and distance close to C2 mean value, the swimmer was categorised in the C1 group. This can be explained by the $DK_{end}$—$LR_{beg}$ relative duration and distance being above the C1 mean in these turns. As illustrated in Fig 3, in C1 and C2, $DK_{end}$—$LR_{beg}$ includes APOend—$AR_{beg}$ [T1], $LR_{beg}$—$AR_{beg}$ [T2], $DK_{end}$—$APO_{beg}$ [T4] and arm pull-out. Due to the small Fisher information, the three time gaps cannot explain the swimmer being characterised in C1. Instead, the primary factor for swimmer 6 switching from C1 to C2 is likely the change in the relative duration and distance during the arm pull-out phase.

Although the absolute 15 m time tended to be slightly faster in C1 than in C2, the 15 m time relative to the 25 m time was 58.04 ± 0.81% and 58.69 ± 0.93% for C1 and C2, respectively. The similarity between these two coordination patterns implies that the swimmers changed their coordination pattern to prevent performance deterioration, which reinforces the functional role of coordination variability rather than the perspective that motion variability is merely biological noise and not beneficial [11].

## Systematic intra-individual variability

There was a significant lap effect on the mean travelling distance and duration of the active underwater phase, which showed that, as a general trend, both variables decreased linearly as the trial progressed. The reduction in the whole underwater phase has been previously reported [16, 27], and the results of the present study confirmed that it was also the case when focusing on the active underwater phase. However, no significant lap effect was observed in the active underwater velocity. These results likely reflected a strategy of elite swimmers reducing the underwater travel distance and duration to maintain the mean underwater velocity [16].

Among the active underwater phase, the swimmers did not change the relative distance and duration of $LR_{beg}$—$AR_{beg}$ [T2] and $AR_{end}$—$LR_{end}$ [T3], but they linearly decreased both the relative distance and duration in APOend—$AR_{beg}$ [T1] and $DK_{end}$—$APO_{beg}$ [T4]. Conversely, the relative duration and distance in $APO_{end}$—$DK_{beg}$ [T5] increased from the first to the last turn. As described earlier, APOend—$AR_{beg}$ [T1] in the present study corresponded to the glide phase with arms alongside the body. Thus, the linear decrease in APOend—$AR_{beg}$ [T1] duration and distance means that the swimmers shortened the glide duration with arms at the side, and consequently, the distance travelled during this period also decreased. As discussed earlier, this might be a strategy of swimmers to maintain the mean active underwater velocity despite the accumulating fatigue, particularly by reducing the non-propulsive phase duration. Fig 3 illustrates that the duration of $APO_{end}$—$DK_{beg}$ [T5] was the sum of the durations of dolphin kick, $DK_{end}$—$APO_{beg}$ [T4] and arm pull-out in the C1 and C2 groups, i.e. all swimmers except for swimmer 10. In the present study, the relative duration of both dolphin kick and arm pull-out increased as the trial progressed, which could explain the increase in the relative duration and distance of $APO_{end}$—$DK_{beg}$ [T5], despite the slight decrease in $DK_{end}$—$APO_{beg}$ [T4] relative duration and distance.

From the first to the last lap, the relative duration of the dolphin kick, arm pull-out and recovery motion increased from 5.14% to 6.21%, 16.9% to 19.9% and 30.9% to 34.2%, respectively. Given that the absolute active underwater phase duration decreased from 3.78 to 3.34 s on average, the absolute change in the duration of each active motion from the first to the last turn was only less than 0.03 s. In contrast, the relative duration of APOend—$AR_{beg}$ [T1] and

$DK_{end}$—$APO_{beg}$ [T4] changed by 5% and 2%, respectively, corresponding to the decrease in the absolute duration of 0.25 and 0.09 s. Therefore, it is reasonable to conclude that the swimmers maintained the duration of dolphin kick, arm pull-out and recovery, while the total duration of the active underwater phase was reduced due to the reduction in $DK_{end}$—$APO_{beg}$ [T4] and the glide duration with the arms at the side ($APO_{end}$—$AR_{beg}$ [T1]). A recent study comparing the inter-limb coordination during the underwater phase in the start segment between 50, 100 and 200 m breaststroke [18] reported that swimmers only adjusted gliding duration but did not change complex inter-limb coordination when performing different events. This is likely due to the glide being the passive motion and least complex movement, whereas active movements such as propulsive and resistive movements require swimmers to efficiently coordinate their four limbs [18]. The results of the present study also confirm that, in general, swimmers adjust their glide motion during the turn segments in 200 m breaststroke. However, it should be noted that this argument cannot be applied to swimmer 10, who performed the dolphin kick after the initiation of arm pull-out and was excluded from the linear mixed modelling for $DK_{end}$—$APO_{beg}$ [T4] and $APO_{end}$—$DK_{beg}$ [T5] as an outlier.

## Summary and limitations

In the present study, Level 2 –Level 4 swimmers [28] were investigated to characterise inter- and intra-individual variabilities in the underwater limb coordination during the turn phase in 200 m breaststroke. The knowledge established in the present study will be particularly useful for further experimental studies that investigate how manipulating swimmers' coordination from one pattern to another will change their performance, which will directly impact coaching and training strategies. Furthermore, future studies should also focus on the inter-limb coordination during the transition stroke by comparing it with that in surface swimming, which will yield a further understanding of the underwater segment in breaststroke swimming.

Three limitations should be noted. Firstly, the present study did not consider the day-to-day variability of the swimmers. Secondly, swimmers performed their 200 m performance without competitors in the present study. Due to these limitations, it should be noted that swimmers who switched their coordination pattern might show slightly different results when being investigated on a different occasion (e.g. they might switch their coordination from one to another at different laps depending on the trial condition). However, such factors also exist in competitions, as it is very unlikely that swimmers compete with exactly the same competitors, in the same pool and lane, and with the same physical conditions. Furthermore, these limitations should not affect the main outcomes of the study (there are three coordination patterns and swimmers generally reduced certain time gaps when the trial progressed) because different conditions could largely affect the swimmers' strategy but likely affect the techniques they use to a lesser extent. Assuming that the systematic results found in the present study could be explained by fatigue, the effect should be present regardless of the condition. Lastly, this study only investigated upper-regional or national-level male adult swimmers. Therefore, caution should be taken when applying the knowledge to novice, international-level, youth, and/or female swimmers.

## Conclusion

Inter-individual variability of the active underwater motion during the 200 m breaststroke was divided into four types: C1, C2, C3, and a combination of C1 and C2. The majority of swimmers (i.e. when they performed with C1 and C2 pattern) completed their dolphin kick before the start of the arm pull-out motion, but both inter-individual, and in some cases, intra-

individual variability were observed in the relative glide (with the arms at the side) duration and distance. In general, swimmers reduced the absolute duration and distance in the active underwater phase throughout the trial due to the decrease in the time gap between the end of the dolphin kick and the start of the arm pull-out and the glide duration with the arms at the side.

## Supporting information

**S1 Dataset.**
(XLSX)

## Author Contributions

**Conceptualization:** Tomohiro Gonjo, Bjørn Harald Olstad, Jan Šťastný, Ana Conceição, Ludovic Seifert.

**Data curation:** Tomohiro Gonjo, Bjørn Harald Olstad.

**Formal analysis:** Tomohiro Gonjo, Bjørn Harald Olstad, Ludovic Seifert.

**Funding acquisition:** Jan Šťastný, Ana Conceição, Ludovic Seifert.

**Investigation:** Tomohiro Gonjo, Bjørn Harald Olstad, Jan Šťastný, Ana Conceição, Ludovic Seifert.

**Methodology:** Tomohiro Gonjo, Bjørn Harald Olstad, Jan Šťastný, Ana Conceição, Ludovic Seifert.

**Project administration:** Ludovic Seifert.

**Software:** Tomohiro Gonjo, Ludovic Seifert.

**Supervision:** Bjørn Harald Olstad, Ludovic Seifert.

**Validation:** Tomohiro Gonjo, Bjørn Harald Olstad, Jan Šťastný, Ana Conceição, Ludovic Seifert.

**Visualization:** Tomohiro Gonjo, Bjørn Harald Olstad.

**Writing – original draft:** Tomohiro Gonjo.

**Writing – review & editing:** Tomohiro Gonjo, Bjørn Harald Olstad, Jan Šťastný, Ana Conceição, Ludovic Seifert.

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
