## [Decision Letter · Decision Letter 0]

24 Aug 2022

PONE-D-22-16543Intra- and inter-individual variability in the underwater pull-out technique in 200 m breaststroke turnsPLOS ONE

Dear Dr. Gonjo,

Thank you for submitting your manuscript to PLOS ONE. After careful consideration, we feel that it has merit but does not fully meet PLOS ONE’s publication criteria as it currently stands. Therefore, we invite you to submit a revised version of the manuscript that addresses the points raised during the review process.

We look forward to receiving your revised manuscript.

Kind regards,

Dalton Müller Pessôa Filho, Ph.D.

Academic Editor

PLOS ONE

Journal Requirements:

 "AC received Portuguese Foundation for Science and Technology, I.P. under Grant UID04045/2020, JS obtained the Institutional developing project under grant RP 902025009 of Brno University of Technology, Czech Republic, and LS was supported by the French National Agency of Research under Grant ANR-19-STHP-0004, NePTUNE project. All funders were not involved in the study design, data collection and analysis, decision to publish or preparation of the manuscript. "

Reviewers' comments:

Reviewer's Responses to Questions

**Comments to the Author**

1. Is the manuscript technically sound, and do the data support the conclusions?

Reviewer #1: Yes

Reviewer #2: Yes

2. Has the statistical analysis been performed appropriately and rigorously? 

Reviewer #1: Yes

Reviewer #2: Yes

3. Have the authors made all data underlying the findings in their manuscript fully available?

Reviewer #1: Yes

Reviewer #2: Yes

4. Is the manuscript presented in an intelligible fashion and written in standard English?

Reviewer #1: Yes

Reviewer #2: Yes

5. Review Comments to the Author

Reviewer #1: GENERAL COMMENT

The present manuscript contains an interesting study about variations on the breaststroke pullout actions throughout a 200m breaststroke time trial. Authors make a thoughtful data analysis and present results in a proper document from a formal point of view. The main concern for the present reviewer is that results and discussion are difficult to follow due to terminology employed. Authors could make an effort to present coordination analysis in a more “intuitive” way. As a former swimming coach with experience in the analysis of breaststroke swimmers, the present manuscript is far from clearly extracting the main practical applications.

More specifically, in relation to time gaps during pull out actions, T1-T6 would be expected to be named in the chronological order they occur. Of course, there are different pull-out combinations between swimmers but literature has described the most common patters both in competition and experimental conditions. For example, T6 would usually occur the first time gap…why is then called T6? Considering all swimmers in the present sample performed dolphin kick before the arm pullout…why both T4 and T5 were included? For the simplicity sake,… could T5 be removed from analysis? (lines 277-283 are an example on how complicated this analysis gets)

A suggestion of time gaps in chronological order would be:

1. Dolphin kick to arm pullout

2. Dolphin kick to leg recovery

3. Arm pullout to arm recovery

4. Arm recovery to Leg recovery

5. Breaststroke kick to Arm stroke

In addition, it would be much more easy-to-understand if time gaps were named according to the key events they refer. For example, the “dolphin kick to arm pullout” or “DK-AP” gap instead of T1. Otherwise, it is difficult for readers to follow arguments in the manuscript even if they are used to the swimming terminology (because there are 6 time gaps, maybe with one or two would be easier).

SPECIFIC COMMENTS

Line 57 is there any evidence on the role of inter-limb coordination in underwater swimming? Not sure the reference of Connaboy 2009 covers this type of coordination during UUS

Line 171, despite previous studies of the same group provided a graphical illustration of pull out phases and time gaps, there is no graphical description in the present study. This undoubtedly would help readers to underwater what authors are measuring. At least, a clear reference to previous studies where a graphical description was included should be cited in the methods section.

Line 197, it would be interesting to include the first breaststroke kick and subsequent arm pull on the analysis of the breaststroke underwater strategies.

Line 237…”C3 differed from C1 and C2 in all these time gaps” what is the basis for authors saying that there were differences in the time gaps? Is there any type of statistical comparison to support this?

Reference 20: invalid citation??

Lines 249-259 what can provide these observations from individual swimmers to the generalization of the present study results?

Figure 3 contains tones of information. It may be split in two figures: one for AUW parameters and pull out phases and other for the time gaps. The same for table 3

At some point of the results section, it would be useful to present average absolute values of distance and times for each pullout subsection. This could serve as a frame of reference for coaches and practitioners.

Lines 303-313. If only four swimmers (less than half) modified their pullout pattern through the 200m event, then the second hypothesis could be partially rejected.

Lines 316-317 why the three coordination patterns were not tagged as in the mentioned previous publications (i.e. glide, continuity and superposition)?

Line 338. “The differences between C1 and C2 were clearer in T1 and T6 than in T4 and T5” but in line 343-344 “T1 (duration of the glide with the

arms at the side) was identical in these two groups”…isn’t this contradictory?

Reference #16 and #28 are the same

Line 351 Intra-individual variability section is probably too long, authors may consider splitting in two parts.

Line 384 there is at least one more precedent in the literature that examined the underwater evolution on a 200m event in 25m pool

Line 394-395 is there any precedent in the literature of a decrease in the glide time with fatigue?

Lines 425-430 is there any limitation of the present study? This could be indicated here. For example, level of the participants (624 fina points) should be considered not even national level according to Ruiz-Navarro et al (2022)

Lines 443-448 what was the role of the co-author AC?

Reviewer #2: General comments

I would like to congratulate the authors for the effort and time spent conducting this study, which aims, mostly, to investigate the intra- and inter-individual variability in arm-leg coordination during the underwater phase of the turn segment in 200 m breaststroke. This is an interesting study, under the scope of the journal and that could provide some practical applications for researchers, coaches and swimmers. The manuscript is well-written, easy to follow and with appropriate data analysis. These are the main strengths of the manuscript. Although the study is interesting, I have some concerns and below are my specific comments for each section.

Abstract

- “LMM showed a decrease…”, please, be more specific (i.e., throughout the 200m trial?)

- Should authors include that three patterns were found, in the conclusion?

Introduction

- Great work in the introduction. This is easy to read and understand the relevance of the study. Nevertheless, I think that it can be reduced to one or two paragraphs to be easier to follow. Moreover, some references should be added to support some sentences (for example, L60-61: 70-71; 117-120)

- L117-123 The authors introduce here the concept of dynamic system theory and it is not clear in what way this could add something relevant to the introduction at this stage. I agree with a “summary” paragraph but struggled to understand this one. Please, this on this.

Methods

- Very nice section. Congratulations. Some minor comments to consider. i) The warm-up was individual but was there any assessment or control? Could 20-30 min of rest after warm-up be too much (according to the literature, this is enough to reduce the warm-up effects)?

- The swimmers competed alone in the swimming pool. I understand this is the appropriate procedure for this. However, this does not “mimic” a competition situation. Could this influence results? Could the real competition setting influence pattern? Please, this should be included in the discussion section/limitation section.

- The authors should change “200m race” to “time-trial”, as this is more appropriate for the testing procedures that were used.

- Each swimmer performed a 200 m time trial once. There is a day-to-day variability that we should acknowledge in the discussion. To be more accurate, I think authors should replicate the 200m time trial and calculate reliability measures. This would support the main findings.

- By using the swimmer's head to determine horizontal velocity instead of hip centre/body centre, could this conditioned results?

- There is no data regarding the times performed during the trial and how different they were from each personal best. This information should be added as it can be important to validate results.

- Why did the authors choose to use those specific covariates? For example, lap as a fixed effect and the participant as a random effect?

Results

- Perhaps I did not understand well, but, why the authors did not include the initial glide in the analysis.

- Figure 3 is important and should be highlighted.

- In my opinion, Figure 4 does not add anything relevant to this study. However, if the authors choose to include this, this should be included in the results and removed from the discussion.

- There is no information about the 200 m times. This should be included in the results.

Discussion

- The authors analysed swimming patterns and found different patterns (n = 3). However, the information about how this influences performance is not available and, in my opinion, is the major gap in this study. There is no information about the performance (total performance or laps) and if there was there any relationship between the higher level swimmers with some specific pattern. I think the authors should improve the practical application of the findings, perhaps by including some discussion/analysis based on this.

- Could the authors provide some reason for the only swimmer that used a completely different pattern?

- Please remove the results from the discussion (Figure 4). The discussion should not provide new results to the manuscript.

- L425, please be careful. I think that the time and FINA points do not correspond to elite-level swimmers.

- Considering your thoughts and these comments, please include limitations in the discussion section.

6. PLOS authors have the option to publish the peer review history of their article (what does this mean?). If published, this will include your full peer review and any attached files.

Reviewer #1: **Yes: **Santiago Veiga

Reviewer #2: No

---

## [Author Response · Author response to Decision Letter 0]

2 Dec 2022

All responses are included in the attached file.

---

## [Decision Letter · Decision Letter 1]

22 Dec 2022

PONE-D-22-16543R1Intra- and inter-individual variability in the underwater pull-out technique in 200 m breaststroke turnsPLOS ONE

Dear Dr. Gonjo,

Thank you for submitting your manuscript to PLOS ONE. After careful consideration, we feel that it has merit but does not fully meet PLOS ONE’s publication criteria as it currently stands. Therefore, we invite you to submit a revised version of the manuscript that addresses the points raised during the review process. Please submit your revised manuscript by Feb 05 2023 11:59PM. If you will need more time than this to complete your revisions, please reply to this message or contact the journal office at plosone@plos.org. Please include the following items when submitting your revised manuscript:A rebuttal letter that responds to each point raised by the academic editor and reviewer(s). You should upload this letter as a separate file labeled 'Response to Reviewers'.A marked-up copy of your manuscript that highlights changes made to the original version. You should upload this as a separate file labeled 'Revised Manuscript with Track Changes'.An unmarked version of your revised paper without tracked changes. You should upload this as a separate file labeled 'Manuscript'.If applicable, we recommend that you deposit your laboratory protocols in protocols.io to enhance the reproducibility of your results. Protocols.io assigns your protocol its own identifier (DOI) so that it can be cited independently in the future. For instructions see: https://journals.plos.org/plosone/s/submission-guidelines#loc-laboratory-protocols. Additionally, PLOS ONE offers an option for publishing peer-reviewed Lab Protocol articles, which describe protocols hosted on protocols.io. Read more information on sharing protocols at https://plos.org/protocols?utm_medium=editorial-email&utm_source=authorletters&utm_campaign=protocols.

We look forward to receiving your revised manuscript.

Kind regards,

Dalton Müller Pessôa Filho, Ph.D.

Academic Editor

PLOS ONE

Journal Requirements:

Additional Editor Comments:

Dear Authors,

The Reviewer #2 is still suggesting few other new adjustments. Please, address these remaining questions in order to proceed with this paper forward.

Reviewers' comments:

Reviewer's Responses to Questions

**Comments to the Author**

1. If the authors have adequately addressed your comments raised in a previous round of review and you feel that this manuscript is now acceptable for publication, you may indicate that here to bypass the “Comments to the Author” section, enter your conflict of interest statement in the “Confidential to Editor” section, and submit your "Accept" recommendation.

Reviewer #1: All comments have been addressed

Reviewer #2: All comments have been addressed

2. Is the manuscript technically sound, and do the data support the conclusions?

Reviewer #1: Yes

Reviewer #2: Yes

3. Has the statistical analysis been performed appropriately and rigorously? 

Reviewer #1: Yes

Reviewer #2: Yes

4. Have the authors made all data underlying the findings in their manuscript fully available?

Reviewer #1: Yes

Reviewer #2: Yes

5. Is the manuscript presented in an intelligible fashion and written in standard English?

Reviewer #1: Yes

Reviewer #2: Yes

6. Review Comments to the Author

Reviewer #1: Authors responded adequately to the majority of points indicated by reviewer in the first revision. However, there are still some points that should be further revised:

- Authors modified the naming of the different time gaps and this undoubtedly will help readers to better understanding data. However, for the present reviewer (and this is only an opinion), it would be easier if key events were named in the order they usually occur during breaststroke pullout. For example, in T4 DKend > APObeg instead of APObeg - DKend

Or in T6 DKend >LRbeg instead of LRbeg – Dkend

- A selection of real swimming frames containing the typical key events and swimmers positioning in the three different clusters would be much more intuitive and clear for readers than figure 3.

- Considering the large extension of results section, could table 2 presented as supplemental material (appendix)?

- Should tables 4 and 5 contain any additional explanation of the statistical effect presented with the F ratio and p-value? Do they refer to the lap effect, right? Why authors do not explicitly indicate it?

- Could be the decrease in the glide time and distance displayed through the 200m turns (like for example in T1) be interpreted in line with decrease in non-propulsive time gaps during breaststroke swimming (ie. Chollet, D., Seifert, L., Leblanc, H., Boulesteix, L., & Carter, M. (2004). Evaluation of arm-leg coordination in flat breaststroke. International journal of sports medicine, 25(07), 486-495) or in 100m freestyle races (Seifert, L., Boulesteix, L., Carter, M., & Chollet, D. (2005). The spatial-temporal and coordinative structures in elite male 100-m front crawl swimmers. International Journal of Sports Medicine, 26(04), 286-293)?

Reviewer #2: Congratulations on the effort, amending the manuscript according to suggestions. I respect the main thoughts of the authors and I should congratulate for the effort and improvement made.

7. PLOS authors have the option to publish the peer review history of their article (what does this mean?). If published, this will include your full peer review and any attached files.

Reviewer #1: **Yes: **Santiago Veiga

Reviewer #2: No

---

## [Decision Letter · Decision Letter 2]

6 Mar 2023

Intra- and inter-individual variability in the underwater pull-out technique in 200 m breaststroke turns

PONE-D-22-16543R2

Dear Dr. Gonjo,

We’re pleased to inform you that your manuscript has been judged scientifically suitable for publication and will be formally accepted for publication once it meets all outstanding technical requirements.

Kind regards,

Dalton Müller Pessôa Filho, Ph.D.

Academic Editor

PLOS ONE

Additional Editor Comments (optional):

Reviewers' comments:

Reviewer's Responses to Questions

**Comments to the Author**

1. If the authors have adequately addressed your comments raised in a previous round of review and you feel that this manuscript is now acceptable for publication, you may indicate that here to bypass the “Comments to the Author” section, enter your conflict of interest statement in the “Confidential to Editor” section, and submit your "Accept" recommendation.

Reviewer #1: All comments have been addressed

2. Is the manuscript technically sound, and do the data support the conclusions?

Reviewer #1: (No Response)

3. Has the statistical analysis been performed appropriately and rigorously? 

Reviewer #1: (No Response)

4. Have the authors made all data underlying the findings in their manuscript fully available?

Reviewer #1: (No Response)

5. Is the manuscript presented in an intelligible fashion and written in standard English?

Reviewer #1: (No Response)

6. Review Comments to the Author

Reviewer #1: Authors have satisfactorily responded to all the reviewer queries and the feeling is that the manuscript is now ready for publication.

7. PLOS authors have the option to publish the peer review history of their article (what does this mean?). If published, this will include your full peer review and any attached files.

Reviewer #1: **Yes: **Santiago Veiga

---

## [Editor Report · Acceptance letter]

13 Mar 2023

PONE-D-22-16543R2 

Intra- and inter-individual variability in the underwater pull-out technique in 200 m breaststroke turns 

Dear Dr. Gonjo:

I'm pleased to inform you that your manuscript has been deemed suitable for publication in PLOS ONE. Congratulations! Your manuscript is now with our production department. 

Kind regards, 

on behalf of

Prof. Dr. Dalton Müller Pessôa Filho 

Academic Editor

PLOS ONE